# GDAP1 Involvement in Mitochondrial Function and Oxidative Stress, Investigated in a Charcot-Marie-Tooth Model of hiPSCs-Derived Motor Neurons

**DOI:** 10.3390/biomedicines9080945

**Published:** 2021-08-02

**Authors:** Federica Miressi, Nesrine Benslimane, Frédéric Favreau, Marion Rassat, Laurence Richard, Sylvie Bourthoumieu, Cécile Laroche, Laurent Magy, Corinne Magdelaine, Franck Sturtz, Anne-Sophie Lia, Pierre-Antoine Faye

**Affiliations:** 1Maintenance Myélinique et Neuropathies Périphériques, EA6309, University of Limoges, F-87000 Limoges, France; nesrine.benslimane@unilim.fr (N.B.); frederic.favreau@unilim.fr (F.F.); marion.rassat@gmail.com (M.R.); laurence.richard@unilim.fr (L.R.); sylvie.bourthoumieu@unilim.fr (S.B.); laurent.magy@unilim.fr (L.M.); corinne.magdelaine@unilim.fr (C.M.); franck.sturtz@unilim.fr (F.S.); anne-sophie.lia@unilim.fr (A.-S.L.); pierre-antoine.faye@unilim.fr (P.-A.F.); 2CHU Limoges, Service de Biochimie et Génétique Moléculaire, F-87000 Limoges, France; 3CHU Limoges, Service de Neurologie, F-87000 Limoges, France; 4CHU Limoges, Service de Cytogénétique, F-87000 Limoges, France; 5CHU Limoges, Service de Pédiatrie, F-87000 Limoges, France; cecile.laroche@chu-limoges.fr; 6CHU Limoges, Centre de Compétence des Maladies Héréditaires du Métabolisme, F-87000 Limoges, France; 7CHU Limoges, Service de Bioinformatique, F-87000 Limoges, France

**Keywords:** CMT disease, GDAP1, hiPSCs, motor neurons, mitochondria

## Abstract

Mutations in the ganglioside-induced differentiation associated protein 1 (*GDAP1*) gene have been associated with demyelinating and axonal forms of Charcot-Marie-Tooth (CMT) disease, the most frequent hereditary peripheral neuropathy in humans. Previous studies reported the prevalent *GDAP1* expression in neural tissues and cells, from animal models. Here, we described the first GDAP1 functional study on human induced-pluripotent stem cells (hiPSCs)-derived motor neurons, obtained from normal subjects and from a CMT2H patient, carrying the *GDAP1* homozygous c.581C>G (p.Ser194*) mutation. At mRNA level, we observed that, in normal subjects, *GDAP1* is mainly expressed in motor neurons, while it is drastically reduced in the patient’s cells containing a premature termination codon (PTC), probably degraded by the nonsense-mediated mRNA decay (NMD) system. Morphological and functional investigations revealed in the CMT patient’s motor neurons a decrease of cell viability associated to lipid dysfunction and oxidative stress development. Mitochondrion is a key organelle in oxidative stress generation, but it is also mainly involved in energetic metabolism. Thus, in the CMT patient’s motor neurons, mitochondrial cristae defects were observed, even if no deficit in ATP production emerged. This cellular model of hiPSCs-derived motor neurons underlines the role of mitochondrion and oxidative stress in CMT disease and paves the way for new treatment evaluation.

## 1. Introduction

Charcot-Marie-Tooth (CMT) disease is a heterogeneous group of sensory-motor disorders belonging to the larger class of genetic neuropathies. With an estimated prevalence of 1:2500, it is considered as the most frequent inherited pathology of the peripheral nervous system. It indifferently affects both sexes, of any geographical origin and age, and it is clinically defined by muscular weakness and atrophy, foot deformities such as pes cavus, and sometimes sensory loss and balance issues [1]. Traditionally, based on electrophysiological studies, demyelinating forms characterized by reduced nerve conduction velocity (NCV) can be distinguished from axonal CMT forms, with preserved NCV values. A third group, identified as intermediate CMT, has been described, ranking between demyelinating and axonal forms in its clinical aspects [2]. According to the associated mode of inheritance of the disease, CMT can be further classified in autosomal, dominant or recessive forms, and X-linked, dominant or recessive forms. More than 80 genes have been identified to be mutated in these different CMT subgroups [1], although the complete duplication of *PMP22* gene, responsible of the so-called CMT1A, remains the main genetic cause of this pathology.

Mutations in *GDAP1* (ganglioside-induced differentiation protein 1) gene have been reported for the first time in 2001 [3,4], and are well known to induce multiple types of Charcot-Marie-Tooth disease. Autosomal recessive mode of inheritance has been observed in demyelinating (CMT4A or AR-CMTde-GDAP1), intermediate (RI-CMTA or AR-CMTin-GDAP1) and axonal (CMT2H or AR-CMTax-GDAP1) forms, while autosomal dominant mutations seem to lead exclusively to axonal CMT (CMT2K or AD-CMTax-GDAP1) [5,6]. *GDAP1*, located on chromosome 8 of the human genome, encodes a 358 aa protein, expressed on the outer mitochondrial membrane of neurons and, at lower levels, of myelinating Schwann cells [7,8,9]. Even if *GDAP1* CMT-inducing mutations have been largely described in their clinical aspects and associated phenotypes, fewer studies have deeply investigated the molecular mechanisms altered in motor neurons and responsible for the neural degeneration.

Three main model organisms have been developed to elucidate GDAP1 role in cellular physiology, through its up- or down- regulation, in Drosophila melanogaster [10,11], its depletion, in mice [12,13], or its mutation in yeasts [14]. Concerning the cellular models, given the inaccessibility of human neurons, all functional studies employed a large amount of alternative strategies, such as primary cultures of murine neurons [9,15], rat Schwann cells [7], and human fibroblasts [16,17,18], but also transfected, or non-transfected, cell lines, such as Cos7, HeLa, SH-SY5Y, N1E-115, HT22 [7,8,9,15,19,20,21,22,23,24]. These existing models have been fundamental to highlight some GDAP1 functions. GDAP1 involvement in mitochondria fission and fusion events has been observed in N1E-115 cells and in a model of transfected Cos7 cells [7,15,19,20], while HT22 cells and human fibroblasts have been fundamental to reveal GDAP1 implication in protection from oxidative [17,22]. In addition, it seems that GDAP1 takes part, also, in regulating Ca^2+^ homeostasis, as shown in SH-SY5Y and transfected HeLa cells [23,24]. More recently, GDAP1 interaction with the trans-Golgi network, and its involvement in autophagy and maturation of lysosomes, have been suggested in SH-SY5Y and HeLa models [25,26]. However, the lack of motor neurons of human origin did not allow demonstration of the potential extrapolation of these investigations in the CMT patients.

Human induced-pluripotent stem cells (hiPSCs), created for the first time in 2006 [27], have become a powerful tool in the exploration of neurological and neuromuscular diseases. The main advantage of hiPSCs is that they can be obtained from an easy-to-take cell types such as fibroblasts, and they can be potentially differentiated in any kind of cell of human body, such as neurons or glial cells. Moreover, it has been shown that they can be generated from unaffected individuals, but also from affected patients [28]. In the case of Charcot-Marie-Tooth disease, models of hiPSCs-derived motor neurons have been established for different forms associated to different genes, such as *NELF*, *MFN2* [29,30], *HSPB1* [30,31], but also hiPSCs-derived Schwann cells presenting the *PMP22* duplication [32]. HiPSCs for *GDAP1* are reported in three studies [28,29,33], and differentiated into motor neurons only once, in our recent publication [28].

Here, we report the first functional study on human hiPSCs-derived motor neurons from a CMT2H patient, carrying the homozygous c.581C>G (p.Ser194*) mutation in *GDAP1*, underlining the role of mitochondria and oxidative stress in this human GDAP1-defective CMT disease.

## 2. Materials and Methods

### 2.1. Subjects

Ethics approval was obtained from the ethic committee of Limoges University Hospital (n°384-2020-40, 10/07/2020), as well as the informed consent of all participants. The study was conducted in accordance with the Declaration of Helsinki. The family of the CMT-patient presented two cases (Figure 1). The propositus, here reported as “patient”, showed first gait disturbances when he was 18 months old. This young boy was characterized by a severe axonal neuropathy, with subacute progression and polyvisceral disorders, leading him to an early death at the age of three. His 5-year-old younger brother developed motor impairment in feet, distal atrophy and abolished deep tendon reflexes associated to mental retardation. Parents, with a first degree of consanguinity, were asymptomatic, as well as the 13-year-old elder brother. In the *GDAP1* gene genetic analyses detected the c.581C>G (p.Ser194*) mutation, homozygous in the patient and his affected brother, and heterozygous in the other family members. No other mutation in CMT- and peripheral neuropathies-associated genes was detected by targeted Next Generation Sequencing (NGS) (see [34] for the detailed protocol). Two control subjects, without any clinical neurological signs, were enrolled in this study: Ctrl-1, a 24-year-old man, and Ctrl-2, a 28-year old woman. Sanger sequencing excluded any *GDAP1* mutation in these controls.

### 2.2. Skin Biopsies and Fibroblasts Cell Culture

Skin biopsies were obtained from patient, Ctrl-1, and Ctrl-2, and incubated in CHANG Medium^®^ D (Irvine Scientific, Santa Ana, CA., USA), with 10% Fetal Bovine Serum (FBS) (Gibco, Thermo Fisher SCIENTIFIC, Waltham, MA, USA). After two weeks, once fibroblasts (FBs) have migrated from the skin fragment and grew in the culture dish, they were isolated using trypsin. In the first three days, fibroblasts were cultivated combining the Chang Medium^®^ D (25%) with the RPMI 1640 medium (75%) (Gibco, Thermo Fisher SCIENTIFIC), supplemented with 10% FBS. Then, Chang medium D was completely replaced by RPMI medium and FBS.

### 2.3. HiPSCs Generation and Characterization

hiPSCs were generated following the iStem (INSERM/UEVE UMR861, AFM, Genopole, Evry, France) protocol. First day, CF-1 Mouse Embryonic Fibroblasts (MEF), Mitomycin-C treated (TebuBio, Le-Perray-en-Yvelines, France), were seeded on gelatin coating (Sigma-Aldrich, Merck, Kenilworth, NJ, USA), at 25,000 cells/cm^2^ density. The second day, 600,000 fibroblasts, from patient and controls, were reprogramed with three plasmids (Plasmid #6 pCXLE-hOCT3/4 shp53-F Addgene, Plasmid #7 pCXLE-hSK Addgene, Plasmid #8 pCXLE-hUL Addgene), through the Nucleofector II device (Amaxa, Lonza, Basel Switzerland). Reprogramed cells were cultured in DMEM+GlutaMAX medium (Gibco, Thermo Fisher SCIENTIFIC), supplemented with 10% FBS, 1% MEM non-essential amino acids (Thermo Fisher SCIENTIFIC), 1% sodium pyruvate (Thermo Fisher SCIENTIFIC), and, at day 1, with 0.1% gentamycin (Thermo Fisher SCIENTIFIC). Culture medium was replaced every day. After 14–21 days, colonies were selected, using a needle, and transferred in new gelatin/MEF coated dishes. HiPSCs colonies grew in a KO-DMEM medium (Gibco, Thermo Fisher SCIENTIFIC), with 20% KnockOut Serum Replacement (Gibco, Thermo Fisher SCIENTIFIC), 1% MEM non-essential amino acids, 1% Glutamine (Gibco, Thermo Fisher SCIENTIFIC), 0.1% β-mercaptoethanol (Gibco, Thermo Fisher SCIENTIFIC), and 0.1% gentamycin. It was replaced every day, extemporaneously supplemented with 20 ng/mL Fibroblast Growth Factor (FGF2) (PeproTech Inc., Rocky Hill, NJ, USA). After hiPSCs amplification, all quality controls were performed (Appendix A).

### 2.4. Motor Neurons Generation and Culture

Differentiation protocol was applied as previously described by Faye et al. [28]. After neural progenitors (NPs) were obtained, they were seeded on poly-L-ornithine/laminine-coated dishes, and cultured in neural media, adding 100  ng/mL Sonic Hedgehog (Shh) (PeproTech Inc.), 5 µM RA (Sigma-Aldrich, Merck), 10 ng/mL BDNF (brain-derived neurotrophic factor) (PeproTech Inc.), 10 ng/mL GDNF (glial cell line-derived neurotrophic factor) (PeproTech Inc.), and 10 ng/mL IGF-1 (insulin-like growth factor-1) (PeproTech Inc.). Then, they were passed every 3 to 4 days, and maintained at high density. In order to generate completely differentiated motor neurons (MNs), NPs were seeded at a density of 20,000 to 30,000 cells/cm^2^, using the same coating and the same culture medium.

### 2.5. RNA Analysis

Total RNA was extracted from fibroblasts, hiPSCs, NPs and MNs of Ctrl-1 and patient, using the miRNeasy Mini kit (QIAGEN^®^, Venlo, The Netherlands). After verifying RNA integrity with the Bioanalyzer 2100 system (Agilent Technologies), it was converted in cDNA with the QuantiTect^®^ Reverse Transcription kit (QIAGEN^®^). For the quantitative PCR (qPCR, or Real-Time PCR), primers were designed between the fifth and the sixth exon of *GDAP1*, and between the fifth and the sixth exon of *TBP* (TATA-Box Binding Protein), chosen as reference gene. Reactions were prepared with the Rotor-Gene SYBR-Green PCR Kit (400) (©QIAGEN) and performed on the Corbett Rotor-Gene 6000 Machine (© QIAGEN). All qPCR reactions were performed four times.

### 2.6. Immunocytochemistry (ICC)

For the immunofluorescence, cells were fixed in 4% paraformaldehyde (PFA) (Sigma-Aldrich, Merck) for 10 min and permeabilized with 0.1% Triton X-100 (Sigma-Aldrich, Merck) for one hour. They were incubated overnight at 4 °C with the primary antibody, prepared in 3% BSA, and then the next day with the secondary antibody for one hour at room temperature. PFA, Triton X-100 and BSA were diluted in Dulbecco’s phosphate-buffered saline 1X (DPBS) (Gibco, Thermo Fisher SCIENTIFIC). Nuclei were stained with 2 µg/mL 4′,6′-diamidino-2-phénylindole dihydrochloride (DAPI) (Sigma-Aldrich, Merck). Images captures were performed using a fluorescence microscope (Leica Microsystems, Wetzlar Germany) or a confocal microscope (Zeiss, Oberkochen, Germany), while their processing and analysis were made with NIS Element BR software, Zen Black and Zen Bleu software, and Image J software. All antibodies’ dilutions and references are reported in Appendix A. PGP9.5, the ubiquitin carboxyl-terminal hydrolase, Tuj-1, the neural-specific β-tubulin III, and the cholyne acetyltransferase (ChAT) enzyme, were chosen as neural markers. The Prolyl 4-hydroxylase subunit-β (P4HB) was chosen as fibroblasts’ marker, as indicated by previous publications [35,36] and manufacturer instructions (OriGene, Rockville, MD, USA).

The chromogenic 3,3′-Diaminobenzidine (DAB) staining was used as complementary ICC method to immunofluorescence, by reason of its high sensitivity. For the DAB staining, MNs were fixed, permeabilized, and incubated with the GDAP1 primary antibody overnight. Next day, the VECTASTAIN^®^ Elite ABC kit (Vector Laboratories, Burlingame, CA, USA) was used for the avidin-biotin/peroxidase detection. The DAB+ chromogen, i.e., the peroxidase substrate solution, was added to induce the formation of the brown precipitate, visualized with a light microscope.

### 2.7. Electron Microscopy

All manipulations for the electron microscopy were performed in Neurology and Anatomic Pathology departments at University Hospital of Limoges. Cells were fixed in 2.5% glutaraldehyde, then incubated 30 min, at RT, in 2% OsO4 (Euromedex, Souffelweyersheim, France). After washing them with distilled water, they were dehydrated 10 min in a series of ethanol dilutions (30%, 50%, 70%, 95%) and three times in 100% ethanol. At the end, they were embedded overnight in Epon 812. Thin blocks were selected and stained with uranyl acetate and lead citrate and examined using a Jeol 1011 electron microscope.

### 2.8. Adenosine Triphosphate (ATP) Quantification

ATP was dosed using CellTiter-Glo^®^ Luminescent Cell Viability Assay kit (Promega, Madison, WI, USA), and the luminescent signal was recorded with the Fluoroskan Ascent^®^FL (Thermo Fisher SCIENTIFIC,) following manufacturer instructions. DAPI staining was used to normalize luminescence’s values to the number of cells. Reactions were performed in triplicate, and experiments were repeated three times.

### 2.9. Succinate Dehydrogenase (Complex II) Activity

Succinate dehydrogenase activity was measured using the Cell Proliferation Kit I (Roche, Basel, Switzerland), following manufacturer conditions. Absorbance of formazan crystals, at 595 nm, was recorded with the Multiskan™ FC Microplate Photometer (Thermo Fisher SCIENTIFIC), and normalized to the number of cells, measured with the DAPI staining. Reactions were performed in triplicate, and experiments were repeated three times.

### 2.10. Mitochondrial Superoxide Quantification

Fibroblasts and MNs were analyzed in basal conditions as well as in stressed conditions. Stressed wells were treated two hours with 1 mM H_2_O_2_ solution, prepared in culture medium. After the treatment, 5 µM MitoSOX™ Red mitochondrial superoxide marker (Molecular Probes, Thermo Fisher SCIENTIFIC) was added to the whole of the plate and incubated for 10 min at 37 °C. Fluorescent signal was detected using the Leica DM IRB microscope and normalized to the number of cells, measured with the DAPI staining. Reactions were performed in triplicate, and experiments were repeated three times.

### 2.11. Statistical Analysis

All statistical analyses were performed using the GraphPad Prism 5 software (GraphPad Software, Inc., San Diego, CA, USA). Data were expressed as mean ± SEM (Standard Error of the Mean). They were compared using the nonparametric Mann–Whitney U test; *p* < 0.05 was considered significant.

## 3. Results

### 3.1. Control and CMT2H hiPSCs Efficiently Differentiate into MNs

Fibroblasts of Ctrl-1, Ctrl-2, and the CMT2H patient were reprogramed in hiPSCs. After validating hiPSCs of the three subjects for all quality controls (Appendix A), our differentiation protocol was applied in order to generate NPs first, then MNs (all differentiation steps for Ctrl-1, Ctrl-2, and patient are reported in the Appendix A). Figure 2A shows that all cells were PGP9.5-positive (red) and Tuj-1-positive (green), validating their neuronal profile. Cells expressed also the cholyne acetyltransferase (ChAT) enzyme (green), confirming the cholinergic function of these hiPSCs-derived motor neurons (Figure 2B).

### 3.2. GDAP1 mRNA Is Expressed in Neural Cells of Controls, but It Is Absent in p.Ser194* Cells

In order to investigate GDAP1 functions and GDAP1-associated mechanisms, we evaluated the expression of *GDAP1* mRNA, from fibroblasts to hiPSCs, neural progenitors, and motor neurons, of patient and Ctrl-1 subject. In Ctrl-1, we showed that *GDAP1* is differently expressed in the four cell types (Figure 3, yellow plots). It was clear that *GDAP1* is weakly expressed in fibroblasts (0.05 ± 0.02, normalized to *TBP*), and slightly more expressed in hiPSCs (0.38 ± 0.05; *p* < 0.05). In contrast, *GDAP1* mRNA levels significantly increased in neural cell types, associated with the progression of neural differentiation. In particular, *GDAP1* expression was around 33-fold higher in NPs (1.67 ± 0.21; *p* < 0.05), and 56-fold higher in MNs (2.79 ± 0.47; *p* < 0.05), compared to fibroblasts’ levels. In patient’s cells, the same expression trend was observed in the four cell types, thus *GDAP1* mRNA was nearly absent in patient’s fibroblasts (0.01 ± 0.003), and significantly higher in hiPSCs (0.06 ± 0.005; *p* < 0.05), NPs (0.3 ± 0.03; *p* < 0.05), and MNs (0.33 ± 0.04; *p* < 0.05) (Figure 3, red plots). Nevertheless, for each cell type, except fibroblasts, patient’s mRNA levels of *GDAP1* were drastically reduced compared to Ctrl-1, as confirmed by statistical analysis (*p* < 0.05 hiPSC patient vs. hiPSC Ctrl-1; *p* < 0.05 NP patient vs. NP Ctrl-1; *p* < 0.05 MN patient vs. MN Ctrl-1).

### 3.3. GDAP1 Protein Is Expressed in MNs of Controls and Absent in p.Ser194* MNs

To complete the expression study, we evaluated GDAP1 protein expression on MNs, the cellular type known to express GDAP1, in comparison with fibroblasts. The experiment was performed for the two control subjects and the CMT patient (Figure 4).

For the immunofluorescence analysis, MNs were stained with GDAP1 antibody (red), and Tuj-1 antibody (green), specific of neural cells. As shown in Figure 4A, GDAP1 protein was detected in MNs of Ctrl-1 and Ctrl-2, located in neurons’ cell body. In contrast, no fluorescent red signal was observed in patient’s MNs, suggesting the weak expression of GDAP1 protein (Figure 4A).

In fibroblasts, GDAP1 staining (red) was associated to Prolyl 4-hydroxylase subunit-β antibody (P4HB) (green). As expected, according to the RNA measurement, a weak GDAP1 signal was observed, both in controls’ and patient’s fibroblasts (Figure 4B).

These results were supported by the DAB staining showing the higher expression of GDAP1 protein in MNs of Ctrl-1 compared to patient (Figure 5).

### 3.4. GDAP1 Mutation Impacts Cell Proliferation and Viability of Neural Cells

Neural progenitors are known to rapidly proliferate. However, this proliferation decreases throughout their final differentiation into motor neurons. Among all existing markers, we analyzed proliferation rate of NPs with the Ki-67 staining. It was evaluated at day 1, right after NPs seeding, and at day 4, when neurons have started their differentiation process. As shown in Figure 6A, at d1, about 75% of NPs of Ctrl-1 (75.04 ± 1.83%) and Ctrl-2 (75.45 ± 1.58%) were Ki-67-positive, whereas only 60% of patient’s NPs expressed Ki-67 (60.26 ± 1.60%; *p* < 0.01). At d4, proliferating cells were slightly lower than 60% in Ctr-1 (57.7 ± 1.37%) and Ctrl-2 (59.38 ± 3.69%), and considerably reduced to 40% in patient’s cells (42.23 ± 4.54%; *p* < 0.05) (Figure 6B).

These results were supported by motor neurons viability. Throughout the NPs differentiation, we observed, for patient, a high number of dead cells. Given that, at d0, the same cells’ number had been plated for the three subjects, at day 7, we performed a DAPI staining to evaluate the number of surviving cells. A significant 60% reduction of patient’s MNs viability was revealed, compared to Ctrl-1 (0.45 ± 0.05 vs. 0.95 ± 0.08; *p* < 0.001) or Ctrl-2 (0.52 ± 0.06 vs. 1.12 ± 0.09; *p* < 0.001) (Figure 7).

### 3.5. In MNs, GDAP1 Mutation Is Associated with Cytosolic Lipid Droplets and Perturbed Mitochondrial Morphology

HiPSCs-derived motor neurons of controls and patient were analyzed by electron microscopy. Surprisingly, in the cytoplasm of multiple patient’s MNs, we observed the emergence of several round structures, of various sizes, suspected to be lipid droplets (LDs) (Figure 8F). These structures appeared electron-dense, with a homogeneous content surrounded by a more electron-dense line, presumably a lipid monolayer, supporting their identity as LDs. However, we cannot exclude the possibility of a bilayer, and a different nature of these structures. They were not observed in controls’ MNs, nor in fibroblasts of the three subjects (Figure 8A–E).

Given the mitochondrial localization of GDAP1 protein, we investigated mitochondrial morphology and structure. Looking at MNs’ ultrastructure, any difference in mitochondrial size and shape was remarked between controls and patient. Moreover, MNs of both subjects presented elongated and fragmented mitochondria. However, focusing on mitochondrial cristae, we observed that their organization was altered in mitochondria of patient’s MNs. In particular, cristae’s regular distribution and thickness were perturbed, preventing to discriminate their structure in the internal mitochondrial compartment. Swollen cristae were also observed (Figure 9F). This disorganization of mitochondrial cristae was not present in Ctrl-1 and Ctrl-2 MNs, as well as in fibroblasts of the three subjects (Figure 9A–E).

### 3.6. GDAP1 Mutation Does Not Strongly Alter Oxidative Phosphorylation

The alteration of cristae organization in mitochondria of patient’s MNs, led us to investigate the oxidative phosphorylation through the activity of the electron transport chain (ETC) complexes and the ATP production. Given the limited availability of hiPSCs-derived MNs, we performed the MTT test to evaluate the activity of the succinate dehydrogenase (complex II). In both fibroblasts and motor neurons, succinate dehydrogenase activity seemed to be slightly increased in patient’s cells, compared to Ctrl-1 and Ctrl-2, and it reached significant difference in fibroblasts (Ctrl-2 FB 0.97 ± 0.03 vs. patient FB 1.099 ± 0.03; *p* < 0.05) (Figure 10). In contrast, ATP levels were not significantly different between patient’s and controls’ fibroblasts and motor neurons (Figure 11).

### 3.7. GDAP1 Mutation Could Promote Mitochondrial Oxidative Stress

Mitochondria are also the main producers of reactive oxygen species (ROS), such as superoxide anion, inducing oxidative stress. A perturbation of mitochondrial cristae could promote redox imbalance. As expected, and shown in Figure 12, in patient’s MNs, superoxide anion levels were significantly higher than in Ctrl-1 (Ctrl-1 MN 0.92 ± 0.17 vs. patient MN 1.37 ± 0.15; *p* < 0.05). This significant difference was even observed in fibroblasts (Ctrl-1 FB 0.92 ± 0.17 vs. patient FB 2.61 ± 0.68; *p* < 0.05). The same trend emerged in the comparison with Ctrl-2 cells, but significant difference was not reached.

## 4. Discussion

More than 80 mutations in *GDAP1* gene have already been reported to be responsible for demyelinating, axonal, and intermediate forms of Charcot-Marie-Tooth disease, and associated to heterogeneous phenotypic manifestations [37]. However, *GDAP1* role in cellular functions and processes has not been clearly elucidated, although several relevant studies have been performed [7,13,16,22,38]. Thus, the pathological role of GDAP1 in this disease development remains to be understood. Cellular and animal models expressing *GDAP1* mutations surely represent the most accessible and easier tool to mimic *GDAP1*-induced pathophysiology. Murine and human GDAP1 proteins share 94% of amino acid homology supporting the relevance of this rodent model in expression and localization studies [7,9], structural studies [39], functional studies on knockout animal models [12,13]. Further analyses were conducted using Drosophila, and its *GDAP1*-ortholog gene (*CG4623*) [10,11], or yeast models, transfected with the human *GDAP1* [14,40]. The high intra-species and inter-species variability, together with the complexity of GDAP1 molecular mechanisms involved in Charcot-Marie-Tooth disease, may limit the animal models’ reliability. Concerning the cellular models developed for GDAP1, most of them have animal origin (mice, rats) [7,9,15], or are immortalized cell lines, naturally expressing *GDAP1* (SH-SY5Y, N1E-115, HT22) [7,8,22,24], or by transfection (HeLa, Cos7) [9,15,20,21]. Indeed, a limited number of cell types express *GDAP1*, notably neurons and Schwann cells. These cell types cannot be obtained from humans, and used in vitro as cellular models. Given the inability to culture human neural cells, the only model available in these conditions was represented by human fibroblasts [16,17,18,22], which, unfortunately, poorly express *GDAP1* [22]. The aim of this study was to go beyond limits imposed by existing animal and cellular models, developing a new solid model of human motor neurons, carrying the homozygous p.Ser194* mutation in *GDAP1*, to investigate GDAP1 functions and GDAP1-associated mechanisms in CMT disease.

We first questioned about *GDAP1* expression. In animal models, such as mice and rats, *GDAP1* has been shown to be largely expressed in neurons. In particular, the higher expression was detected in cerebellum, cerebral cortex, hippocampus, olfactory bulb, spinal nerve, but also in sciatic nerve, and motor and sensory neurons [7,8,9]. *GDAP1* expression in Schwann cells was controversial, whereas, in non-neural tissues, it was poorly explored [7,8,9]. Here, we compared, for the first time, *GDAP1* mRNA expression in four human cell types of the same control subject (Ctrl-1): fibroblasts, hiPSCs, NPs, and MNs. Our study revealed, on Ctrl-1 cells, that *GDAP1* is weakly expressed in fibroblasts and hiPSCs, while its expression was significantly higher in NPs, and, above all, in MNs. It is interesting to note that *GDAP1* mRNA in fibroblasts represented only 3% of NPs-*GDAP1* mRNA, and 1.8% of MNs-*GDAP1* mRNA. This is in agreement with Noack et al. work, who showed that control human fibroblasts expressed only 2.6% of *GDAP1* mRNA compared to embryonic stem cells-derived motor neurons [22]. In contrast, in patient’s cells, presenting the homozygous codon-stop mutation c.581C>G in exon 5, *GDAP1* mRNA was only 10–20% of mRNA estimated in each cell type of Ctrl-1, up to 6- and 8-fold smaller than those assessed in Ctrl-1 NPs and Ctrl-1 MNs. Our data seem to suggest that *GDAP1* mRNA is degraded in patient’s cells. As mutated *GDAP1* mRNA contains a premature termination codon (PTC), the nonsense-mediated mRNA decay (NMD) system could be activated and induce its degradation, preventing the synthesis of a truncated, and maybe non-functional, protein [41]. In any case, the NMD system is not always 100% efficient and some PTC-mRNA can escape NMD, and be detected by qPCR, as shown here. Real time qPCR results were also supported by GDAP1 protein expression. Indeed, GDAP1 was not express in fibroblasts, both in controls and patient, while the GDAP1 staining was present in Ctrl’s MNs, and, as expected, lacked in patient’s MNs. Given the high *GDAP1* neural expression, we chose MNs and NPs, as cellular models to investigate its functions, and evaluate its role in Charcot-Marie-Tooth disease development.

However, the weak *GDAP1* expression detected in fibroblasts does not exclude a GDAP1 role in this cell type, and the possibility of conducting functional studies on them [16,17,18]. In our case, in the examination of electron transport chain’s (ETC) activity, any big difference emerged between controls and patient, neither in fibroblasts, nor in MNs. Only a slight increase in succinate dehydrogenase activity was observed, but did not reach significant levels in MNs, and ATP production was preserved in patient’s fibroblasts and MNs. Nevertheless, we cannot exclude that markers used to evaluate oxidative phosphorylation can be limited, and the analysis not-exhaustive. Thus, the measurement of each complex activity, by oxygraphy, could prove more comprehensive in investigation of ETC function.

On the other hand, morphological and oxidative stress analyses, allowed highlighting, exclusively in MNs, two main mechanisms which could play a key role in CMT disease progression: the deregulation of mitochondria morphology and dynamics, and the redox imbalance. These aspects support the relevance of our MNs cellular model in the study of *GDAP1*-associated pathophysiology. First, we investigated mitochondrial morphology. Electron microscopy revealed, only in patient’s MNs, a general disorganization of mitochondrial cristae, which could affect the inner mitochondrial space and subsequent metabolism. The disruption of cristae structure has already been associated to other pathological conditions induced by mutations or lack of proteins involved in mitochondrial dynamics, such as optic atrophy 1 (OPA1) protein [42], or mitofusin 2 (Mfn2) protein [43]. Cristae abnormalities were also reported in nerves’ axonal mitochondria of a CMT2 patient, carrying the c.174_176delGCCinsTGTG (p.Pro59Valfs*4) mutation in *GDAP1* [44], but also, more recently, in muscular tissue of a patient carrying the c.77T>G (p.Leu26Arg) and the c.505_511del (p.Ser169*) *GDAP1* mutations [18].

Interestingly, some additional findings of our cell culture need to be pointed out. We observed in patient’s hiPSCs carrying the p.Ser194* mutation a higher spontaneous differentiation and a reduced maintenance of stemness compared to controls’ hiPSCs, where *GDAP1* seems to be expressed, even if at a low level. Moreover, we have demonstrated that patient’s neural cells present a lower proliferation rate, compared to controls. Both these aspects could potentially be related to GDAP1 involvement in mitochondrial dynamics. Indeed, it is now well known that the proper preservation of mitochondrial dynamics is a fundamental condition for cell cycle progression, and fragmentation of mitochondrial network is required during the mitosis phase [45]. Thus, the deficiency of a fission protein, such as GDAP1, could impact the regulation of cell proliferation mechanisms. Indeed, Prieto et al., demonstrated that *GDAP1* knockout, altering the fission machinery, impairs OSKM (Oct4/Klf4/Sox2/cMyc) reprogramming, and cell cycle progression, in murine iPSCs [46]. Based on our preliminary results and previous studies, we can assume that GDAP1 protein may be a key component in controlling mitochondrial morphology and dynamics, and its lack may disturb mitochondrial-dependent processes, such as in different cell types.

In the cytoplasm of patient’s MNs, round electron-dense structures were observed, suspected to be lipid droplets. LDs could be considered as an accumulation of energetic substrate, such as triglycerides, linked to a defect of mitochondrial beta oxidation, or a hallmark of cellular stress, previously observed in nutrient imbalance, inflammation and oxidative stress [47]. Moreover, several studies have demonstrated that their accumulation is one of earliest events following the induction of cellular apoptosis [48]. Thus, the accumulation of LDs could also be considered as an early signal of apoptotic pathways’ activation, explaining the significant reduction of patient’s MNs observed in last steps of neural differentiation. The synthesis of lipid droplets in neurons has been observed in pathogenesis of several neurodegenerative diseases, such as amyotrophic lateral sclerosis, Huntington’s disease, Alzheimer’s disease, Parkinson’s disease and Hereditary spastic paraplegia [49]. LDs were also described in the ultrastructural analysis of motor neurons obtained from *GDAP1* knockout mice [13], in accordance with our results, corroborating the consistency of our cellular model. In stress conditions, a cytoprotective role against reactive oxygen species (ROS) is also supplied by LDs. As polyunsaturated fatty acids (PUFAs) are more susceptible to ROS-induced peroxidation when they are integrated in cellular membranes, LDs sequester them as triacylglycerols (TAGs) in their core, protecting cellular structures from ROS damage [50]. This phenomenon could probably be responsible for the LDs formation in our cellular model of CMT-motor neurons carrying the *GDAP1* p.Ser194* mutation. In fact, *GDAP1* has been suggested to have an antioxidant role in cellular homeostasis [10,22]. Consequently, in patient’s MNs, the lack of GDAP1 protein increases the amount of generated ROS, proven, in our study, by the MitoSOX™ Red mitochondrial superoxide indicator analysis. However, we have also detected a significant increase of ROS in patient’s fibroblasts, where *GDAP1* is weakly expressed and LDs lacking. The overproduction of superoxide anion, in *GDAP1*-mutated fibroblasts, has also been reported in a recent work [17]. Moreover, the same study demonstrated that *GDAP1*-mutated fibroblasts presented also a reduced expression of Sirtuin 1 (SIRT1) enzyme, which activates the PPARgamma coactivator-1alpha (PGC1 α) [17]. PGC1 α is a fundamental factor in mitochondrial biogenesis, and it has been associated to neurological disorders and diabetic peripheral neuropathy [51,52,53,54]. These data could strengthen the idea that GDAP1 protein, even if at lower levels, could also be present in cell types other than neural cells. In this case, nevertheless, other molecular mechanisms and proteins would take part to the cellular antioxidant defense, counterbalancing GDAP1 deficiency.

## 5. Conclusions

In conclusion, the role of *GDAP1* impairment in Charcot-Marie-Tooth pathophysiology through mitochondrial dysfunction and oxidative stress development was underlined in an original human model of motor neuron from patient’s fibroblasts, carrying the homozygous codon-stop c.581C>G (p.Ser194*) mutation. The results underlined that *GDAP1* is mostly expressed in neural cell types such as MNs and PNs, but also, at lower levels, in fibroblasts and hiPS cells. In patient’s cells, 80–90% of *GDAP1* mRNA would be degraded by the NMD system, leading to the considerable reduction of GDAP1 protein. Taken together, these results demonstrated that hiPS cells can be a powerful tool to recreate any suitable cellular model from patients carrying mutations and are essential for understanding the pathophysiological role of the altered protein, but also necessary to develop new therapeutic strategies.

## Figures and Tables

**Figure 1 biomedicines-09-00945-f001:**
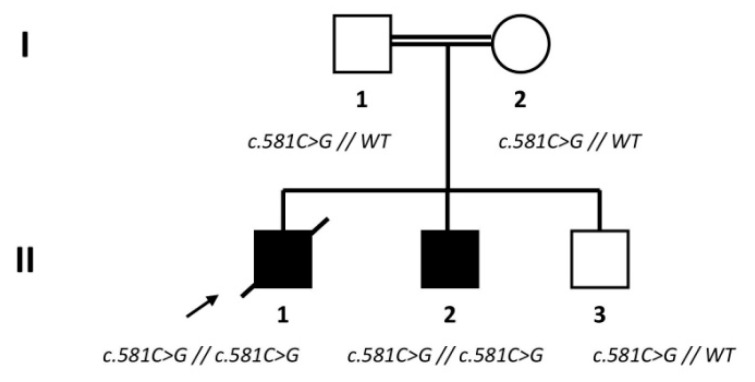
Pedigree of patient’s family with *GDAP1* genotype; the propositus (“patient”) is indicated by the black arrow (WT: wildtype).

**Figure 2 biomedicines-09-00945-f002:**
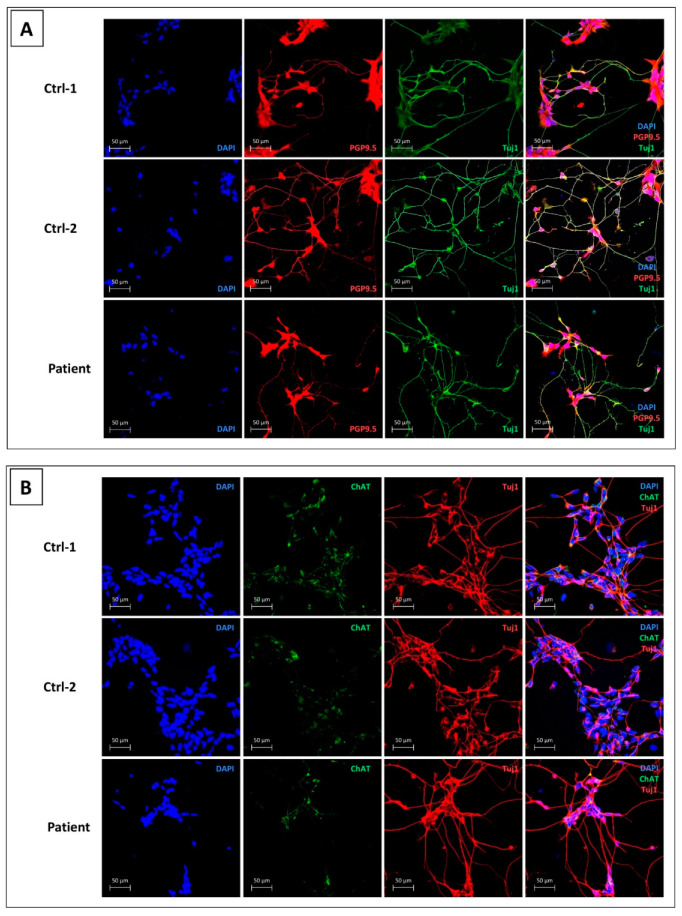
Characterization of hiPSCs-derived motor neurons, for Ctrl-1, Ctrl-2, and patient. (**A**) DAPI (blue) for nuclei staining, PGP9.5 (red), and β-tubulin III (Tuj1) (green); (**B**) DAPI (blue) for nuclei staining, choline acetyltransferase (ChAT) (green) and β-tubulin III (Tuj1) (red). Scale bar = 50 µm.

**Figure 3 biomedicines-09-00945-f003:**
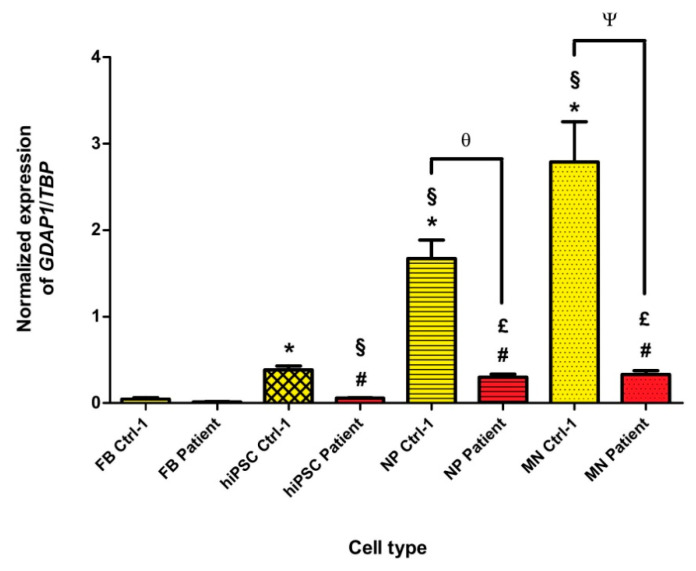
Normalized *GDAP1* expression in fibroblasts (FBs), human induced-pluripotent stem cells (hiPSCs), neural progenitors (NPs), and motor neurons (MNs), from Ctrl-1 subject (yellow plots) and CMT patient (red plots). *TBP* was chosen as reference gene. * *p* < 0.05 vs. FB Ctrl-1; § *p* < 0.05 vs. hiPSC Ctrl-1; # *p* < 0.05 vs. FB patient; £ *p* < 0.05 vs. hiPSC patient; θ *p* < 0.05 vs. NP Ctrl-1; Ψ *p* < 0.05 vs. MN Ctrl-1.

**Figure 4 biomedicines-09-00945-f004:**
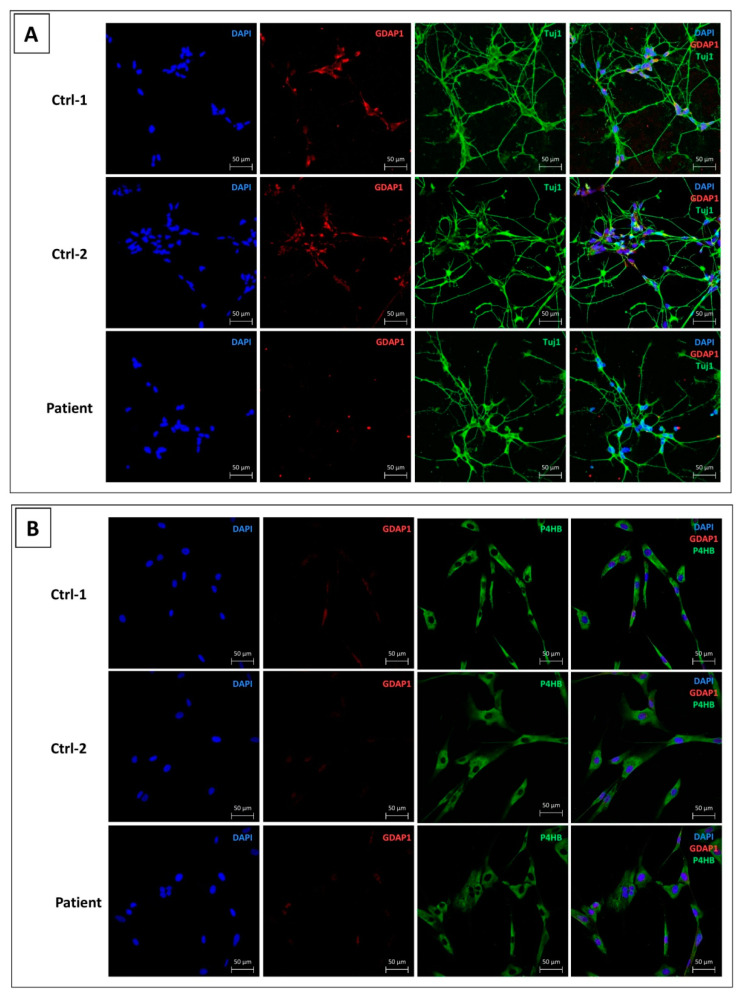
GDAP1 protein expression in MNs (**A**) and fibroblasts (**B**). (**A**) DAPI (blue) for nuclei staining, GDAP1 (red), and β-tubulin III (green), as neural marker. (**B**) DAPI (blue) for nuclei staining, GDAP1 (red), and prolyl 4-hydroxylase subunit-β antibody (P4HB) (green), as fibroblast marker. Scale bar = 50 µm.

**Figure 5 biomedicines-09-00945-f005:**
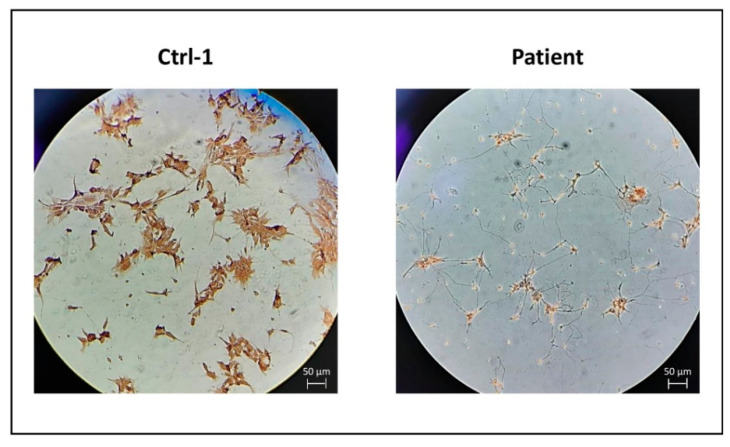
DAB staining for GDAP1, in MNs from Ctrl-1 and patient. Scale bar = 50 µm.

**Figure 6 biomedicines-09-00945-f006:**
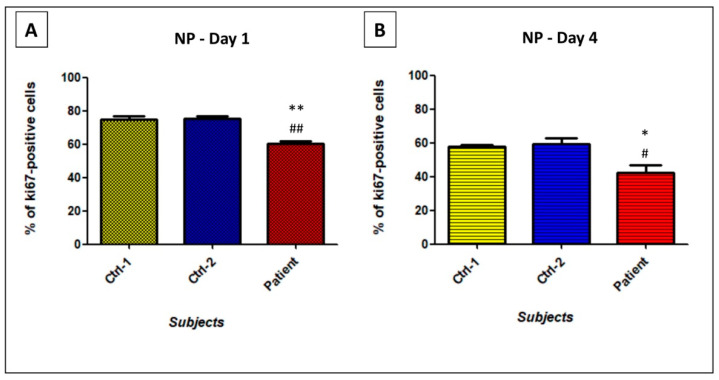
Percentage of Ki-67 expressing-NPs of Ctrl-1, Ctrl-2, and Patient, at day 1 (**A**) and day 4 (**B**) after the beginning of the differentiation process. Lower growth rate in patient’s neural cells, carrying the p.Ser194* mutation in *GDAP1*, was observed. ** *p* < 0.01 vs. Ctrl-1; ## *p* < 0.01 vs. Ctrl-2; * *p* < 0.05 vs. Ctrl-1; # *p* < 0.05 vs. Ctrl-2.

**Figure 7 biomedicines-09-00945-f007:**
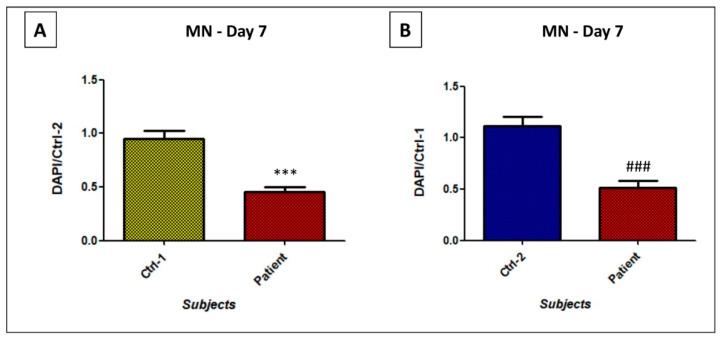
DAPI staining on MNs from patient vs. Ctrl-1 (**A**), and Ctrl-2 (**B**), at day 7 after the beginning of the differentiation process. *** *p* < 0.001 vs. Ctrl-1; ### *p* < 0.001 vs. Ctrl-2.

**Figure 8 biomedicines-09-00945-f008:**
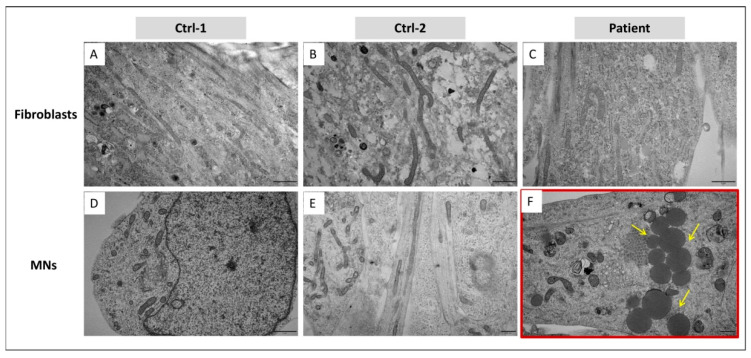
Ultrastructure analysis, by EM, on fibroblasts of Ctrl-1 (**A**), Ctrl-2 (**B**), and patient (**C**), and MNs of Ctrl-1 (**D**), Ctrl-2 (**E**), and patient (**F**). Lipid droplets (LDs), in patient’s MNs, are indicated by yellow arrows. Scale bar = 1 µm in (**A**–**D**); Scale bar = 500 nm in (**E**,**F**).

**Figure 9 biomedicines-09-00945-f009:**
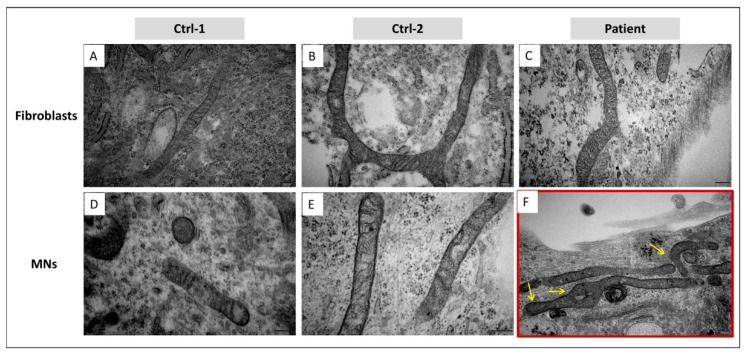
Ultrastructure analysis, by EM on ultrathin sections, of mitochondria infibroblasts of Ctrl-1 (**A**), Ctrl-2 (**B**), and patient (**C**), and MNs of Ctrl-1 (**D**), Ctrl-2 (**E**), and patient (**F**). Yellow arrows point out the alteration of patient’s MNs’ mitochondrial cristae, exhibiting a perturbation of distribution and thickness. Scale bar = 200 nm in (**A**–**F**).

**Figure 10 biomedicines-09-00945-f010:**
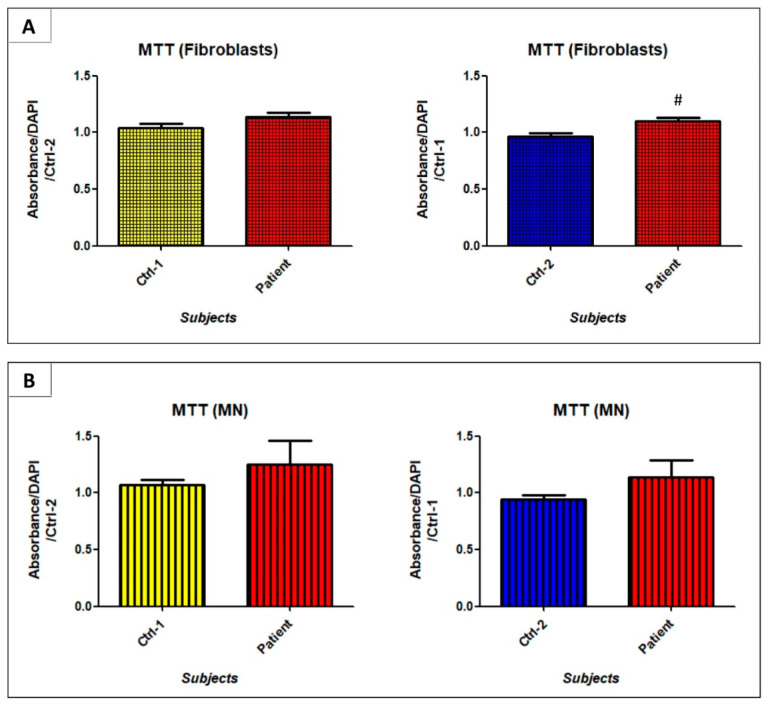
Succinate dehydrogenase activity evaluated in fibroblasts (**A**) and MNs (**B**) of Ctrl-1, Ctrl-2, and patient. # *p* < 0.05 vs. Ctrl-2.

**Figure 11 biomedicines-09-00945-f011:**
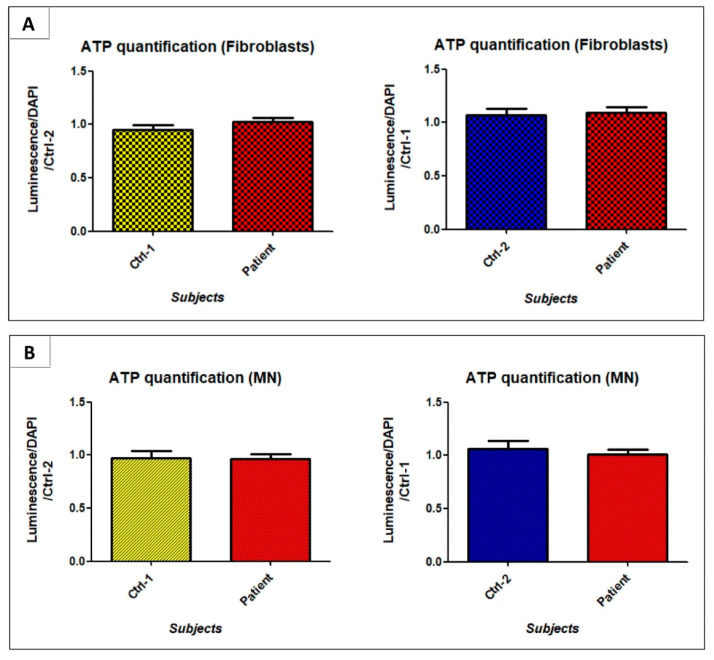
ATP production evaluated in fibroblasts (**A**) and MNs (**B**) of Ctrl-1, Ctrl-2, and patient.

**Figure 12 biomedicines-09-00945-f012:**
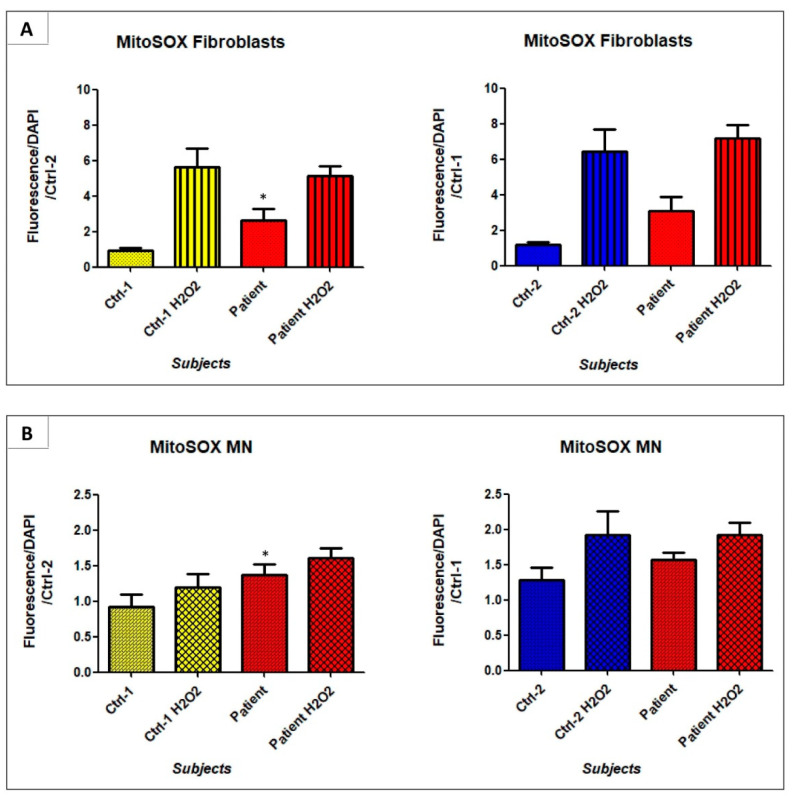
Superoxide anion quantification in fibroblasts (**A**) and MNs (**B**) of Ctrl-1, Ctrl-2, and patient, using the MitoSOX™ Red mitochondrial superoxide indicator. Cells treated with 1 mM H_2_O_2_, 2 h at 37 °C (fibroblasts and MN) were used as positive controls. * *p* < 0.05 vs. Ctrl-1.

## Data Availability

The data presented in this study are available on request from the corresponding author.

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
