# Peer review of "GDAP1 Involvement in Mitochondrial Function and Oxidative Stress, Investigated in a Charcot-Marie-Tooth Model of hiPSCs-Derived Motor Neurons"

_biomedicines, 2021, doi:10.3390/biomedicines9080945_

Round 1

Reviewer 1 Report

In this study authors focus on Charcot-Marie-Tooth (CMT) disease. They generated human induced-pluripotent stem cells (hiPSCs)-derived motor neurons, obtained from normal subjects and from a CMT2H patient, carrying the GDAP1 homozygous c.581C>G (p.Ser194*) mutation. By morphological and functional experiments authors showed, in CMT patient’s motor neurons, a decrease of cell viability associated to lipid dysfunction and mitochondrial distress. Importantly, mitochondrial cristae defect was also observed by ultrastructure analysis.

This manuscript provides novelties and describe a cell-based tool for in vitro investigations in the field of CMT disease. Their conclusions are adequately supported by the data.

However, I would like to suggest some minor concerns:

It would be more helpful if a western blot analysis can be added to imaging experiments.

  1. Protein extracts from fibroblasts, hiPSCs, NPs and MNs, of Ctrl-1 and patient should be tested by western blot analysis in order to compare expression levels of mitochondrial markers, such as TOMM20, COXIV and HSP60.
  2. In addition, PCG1alpha should be checked to evaluate the mitochondria biogenesis.
                1.  

Reviewer 2 Report

In this interesting work, Miressi and collaborators use hiPSCs (Human induced-pluripotent stem cells)-derived motor neurons from a Charcot-Marie-Tooth pacient (CMT2H), with a homozygous mutation in the GDAP1 (ganglioside-induced differentiation protein 1) gene, to analyze the involvement of this gene in the proliferation and viability of neural cells, and in mitochondrial function and oxidative stress. The authors show that GDAP1 expression is drastically reduced in the cell types analyzed of the patient, and that the mutation affects the viability of motor neurons and gives rise to cytosolic lipid droplets and altered mitochondrial morphology, which is reflected in higher levels of oxidative stress. The results are consistent, and the findings make a useful contribution to the literature which is appropriate for Biomedicines. I have only a few comments about some aspects of the manuscript that I think could be clarified and/or improved.

- Figure 5: I have several questions related to this figure. First, why is it not converted into Figure 4, (and current 4 to 5) if it appears first in the text? Second, I think this figure is nor properly introduced. Here the authors evaluate GDAP1 expression using immunohistochemistry with DAB, but this is not explained at the beginning of the section, but at the end, where it is only mentioned that immunofluorescence supports the DAB results. I suggest inserting a few lines when first mentioning the figure describing the experiment more in detail. Also, why were both immunohistochemistry and immunofluorescence performed? Is it to account for possible differences due to sensitivity?

- Figure 5: Also in this figure, are both images taken with the same magnification? I have the feeling that the left image (control) was taken with higher magnification that the right one (patient). This makes it hard to compare the expression levels. Also, in line 258 the authors state that “no fluorescent red signal was observed in patient’s MNs, suggesting the absence, or weak expression”. I think that, looking at the DAB staining, absence is not correct, but just weak expression.

- Materials and Methods, lines 165-169: The authors indicate several concentrations (4% paraformaldehyde, 0.1% Triton X-100, 3% BSA, etc.) without indicating in what they are diluted (PB? PBS?).

- Materials and Methods, lines 175-176: I suggest including a reference to support this sentence.

- Materials and Methods, line 189: examined.

- Results, line 222: Level does not sound correct here to me. Profile maybe?

- Results, line 284: surviving cells.

- Results, line 298: in the cytoplasm.

- Results, line 301: The authors describe these structures as electron-dense, with homogeneous content and surrounded by a more electron-dense line. As described, it sounds as if they are surrounded by monolayer, supporting their identity as lipid droplets. However, if they are surrounded by a bilayer, these structures could be emerging lysosomes or, more unlikely, peroxisomes. In figure 8 it is impossible to see the type of membrane they have. Could the authors incorporate a more detailed image to discern the type of membrane and discuss whether it is bilayer or monolayer in the text?

- Figure 9F: I suggest incorporating a more detailed image to better appreciate the morphology of the cristae.

- Discussion, line 351: I do not understand the meaning of this sentence. Maybe rephrase to “GDAP1 role in cellular functions and processes has not been clearly elucidated, although several relevant studies have been performed”.

- Discussion, line 403: “…but did not reach…”.

- Discussion line 429: change It to it

- Discussion, line 439: in the cytoplasm
